# A Single-Case Series Trial of Emotion-Regulation and Relationship Safety Intervention for Youth Affected by Sexual Exploitation

Jessica J. Laird [1,*], Bianca Klettke [1,2], Sophie Mattingley [1], David J. Hallford [1] and Kate Hall [1,3]

1   School of Psychology, Faculty of Health, Deakin University, Geelong, VIC 3220, Australia;
    bianca.klettke@deakin.edu.au (B.K.); smatting@deakin.edu.au (S.M.); david.hallford@deakin.edu.au (D.J.H.);
    kate.hall@deakin.edu.au (K.H.)
2   Center for Social and Early Emotional Development, Faculty of Health, Deakin University,
    Melbourne, VIC 3125, Australia
3   Deakin University Centre for Drug Use, Addictive and Anti-Social Behaviour Research, Faculty of Health,
    Deakin University, Melbourne, VIC 3125, Australia
*   Correspondence: jjlaird@deakin.edu.au; Tel.: +61-3-9244-6207

**Abstract:** Child sexual exploitation (CSE) is a significant global problem. Interventions implemented with youth affected by CSE frequently target singular adjacent issues (e.g., substance misuse or running away); however, research indicates these interventions are most efficacious when they simultaneously treat CSE sequelae (e.g., emotion dysregulation) paired with relationship skill-building; yet few such interventions exist. Furthermore, the evidence-based reports on CSE research currently lacks rigorous research methods, such as the use of validated measures and the provision of robust outcome data. The current study aimed to implement a combined emotion regulation and safe-relationships intervention (ERIC + YR: emotion regulation, impulse control and 'your relationships') in a community service providing outreach for young women affected by CSE. A randomised single-case series design was used to test the effects of ERIC + YR on emotion regulation strategies, psychological wellbeing, relationship safety knowledge and behaviours, across repeated measurements for young women affected by CSE ($N = 2$; $M_{age} = 18.00$). Phase A consisted of baseline measures for two to three weeks. Phase B consisted of 8-sessions of ERIC + YR delivered across three to six weeks by practitioners who had undertaken ERIC + YR training. Data collection included pre/post intervention measures as well as a daily questionnaire delivered via a smartphone application. While results showed clinically significant and reliable improvements in psychological wellbeing, no other outcome measures showed change between pre and post-intervention. The current study contributes to the evidence-base as an initial step in illuminating how an empirically driven intervention can be delivered as an adjunctive treatment for youth affected by CSE. Implications inform the current evidence-base, with future directions for intervention research discussed.

**Keywords:** sexual exploitation; child; intervention

## 1. Introduction

Child Sexual Exploitation (CSE) is a growing and dangerous type of sexual violence which leaves children, young people and their greater communities suffering multiple co-occurring adverse outcomes, including trauma, mental illness, emotion dysregulation and substance misuse [1–4]. When young people and children affected by CSE remain unsupported and untreated, they are more likely to experience cycles of ongoing sexual violence throughout their lifetimes compared to those unaffected by CSE [1]. Research indicates effective CSE interventions focus on treating co-occurring issues (e.g., running away and/or emotion dysregulation; [2,3] however, outcomes improve significantly when programs are delivered alongside interpersonal skills [2]. To the authors' knowledge, no

such interventions combining these different components exist for individuals affected by CSE. Furthermore, the majority of CSE intervention research lacks rigorous research design, validated measures and provides limited outcome data [2,3]. Therefore, the current study aims to extend research in this area by piloting a novel intervention targeting emotion regulation and relationship safety for young people who have experienced CSE, via a randomized single-case series design.

## 2. Background

CSE is a complex form of sexual violence that affects children and young people, considered an abusive act where an individual or group takes advantage of a power imbalance to use, force, coerce and/or deceive a child or young person into completing or attempting sexual activity, on or off-line; (a) by an offer or actual exchange of unmet needs or wants of the child/young person (e.g., food, clothing, shelter, substances, money, protection, belonging, affection and/or developmental needs or anything of perceived value to the young person or child); and/or (b) for the economic or social advantage of the perpetrator or facilitator; and (c) irrespective of consent or who initiates or solicitations the contact (e.g., child/young person or perpetrator, adult or peer: [4]. In child abuse research the term 'child' or 'young person' commonly refers to individuals below the legal age of adulthood or otherwise considered by societal norms to be a child [5]. Complexities exist around CSE relating to definitional discrepancies (e.g., juvenile prostitution versus trafficking), consent (e.g., youth and children may be sexually exploited even if the activity appears consensual), grooming (e.g., CSE does not always involve direct physical contact), and technology (e.g., distribution of sexual images or video) [6]. Prior to 2009, sexually exploited children and adolescents were not yet recognized as victims of abuse but were perceived to be involved in prostitution [7]. Therefore, most sexual exploitation research focuses on commercialized or street-based sex work [8] leaving significant gaps in our understanding of how to intervene therapeutically with young people affected by this type of sexual violence.

CSE is reported to occur across a broad range of different forms and sociocultural contexts [9]. In developing countries, CSE is often depicted in the form of 'sex trafficking', and/or as a result of extreme poverty (e.g., forcing families to sell children in exchange for money, or under the premise of employment [10,11]. In contrast, developed countries present another profile of CSE, where gangs and pedophile rings exploit urban children and adolescents for money, or perpetrators prey upon children in foster care systems [12]. While urbanization offers benefits for young people (e.g., increased support services and employment opportunities) there are disadvantages, including increased costs of housing and food [13]. Higher costs of living have been associated with economic and social vulnerability which may increase young people's experience of homelessness and the likelihood of exposure to CSE [9]. Furthermore, in a post COVID-19 era, research reports growth in the digital sex market [14,15], with online platforms such as OnlyFans or Suicide Girls distributing and profiting from pornographic content and live-streaming, with content including minors [15–17]. Sexual exploitation both online and offline has been associated with similar adverse outcomes, including depression, substance addiction and antisocial behaviors [9,18].

Prevalence data pertaining to CSE is limited and needs improvement; however, research suggests this type of sexual violence affects up to 5% of the general child and youth population worldwide [3,4]. Amongst Western high school populations, previous research indicates that 2.8% of adolescents attending high schools have sold sex to an adult, on or offline [8]; and a 2021 prevalence study in the United Kingdom reporting over half of the participants were approached sexually by an adult during childhood ($n = 121$), of whom one-quarter were sexually exploited ($n = 56$) (Alderson 2021). Similarly, a 2015 study found that 47% of a university student sample had been approached by an adult in a sexual manner when they were under the age of 16, with a fifth of these resulting in CSE [19]. The COVID-19 pandemic and the social isolation measures in response to it are reportedly

correlated with an upward trend in cases of online child sexual exploitation [14]; this is re-iterated by both developing countries (e.g., 20% increase in CSE online material in 2020 in Cambodia [20]) and developed (e.g., Australian statistics show 33,114 reports of CSE online material in 2021, compared to 17,400 in 2018 [14]. With 62% of the world's total population using the internet, and with the scale, complexity, and danger of CSE escalating over time [21], efficacious intervention is imperative.

Sexual violence against children and young people is undeniably a serious human-rights concern, propelling policy, and research to develop efficacious intervention [2]. Despite research indicating a need for intervention, young people and children exposed to CSE may be more reticent to engage with therapeutic supports or complete intervention programs [10]. This reticence is reported to be impacted by entrenched cycles of repeated victimization and self-blame [9], which is associated with reduced disclosures and mistrust of helping services [22]. Compounding cycles of re-victimization, the literature indicates between 70% to 80% of young people exposed to CSE have pre-existing histories of childhood trauma, ranging from abuse, neglect, exposure to violence and parental substance use and/or mental illness [9,23,24]. Additionally, longitudinal data reports individuals affected by CSE commonly experience several co-occurring psychosocial and mental health difficulties, including deliberate self-harm, high-risk sexualized behaviours, substance misuse [9], antisocial behaviours, sex as a form of self-injury, re-victimization in later life [8], and clinically relevant psychopathology (e.g., mood and anxiety disorders: [25,26]. Intervening effectively and early with young people affected by CSE can prevent a trajectory of further psychological and psychosocial harm, specifically including a reduction in running away [27,28], improving self-esteem and coping strategies [29], and minimizing further victimization and trauma [1]. However, research reports delayed and non-disclosure of CSE can prevent young people from accessing early intervention [30,31]. Evidence suggests stigmatization of child sexual abuse is associated with feelings of shame which interferes with the acceptance of help [30,31]. The normalization of violence and unequal power dynamics also impact sociocultural perceptions of CSE, impeding both professional identification of CSE and understanding of CSE within survivors [30,31]. Additionally, there is consensus in the current evidence-base that many children and young people affected by sexual abuse never disclose, and while there may be evidence of abuse (e.g., medical evidence or perpetrator confessions), up to 43% of these young people will still be unwilling and/or unable to disclose [32]. Nondisclosure and delayed disclosure may contribute to longitudinal findings which indicate by the time sexually exploited youth are supported by services, they often report being exposed to long-standing or repeated experiences of sexual victimization [9]. Therefore, by the time a young person or child accesses support, and beyond the scope of early intervention or harm minimization, they likely require intervention level care to provide pathways of restoration and healing from sexual violence.

### 2.1. CSE Intervention

There has been an increase in the development of dedicated CSE preventive strategies around the world (e.g., forming alliances with policing, research, and community services, to raise awareness of CSE, reduce stigma and stop sexual exploitation [33]; however, a recent systematic review indicates the majority of published CSE intervention research lacks statistical rigor, uses outcome measures that are heterogenous and difficult to compare across studies, or lacks data reporting altogether [3,34]. Given these limitations, the study of evidence-based interventions for CSE prevention and intervention can be considered to still be in its infancy.

Evidence regarding interventions that target co-occurring psychosocial issues in those affected by CSE (e.g., with a focus on distal factors such as school disengagement or juvenile justice) reports large variability across the efficacy and type of intervention utilized [2]. Research indicates that skills-based, trauma-focused, and cognitive behavioral therapy were particularly effective in treating young people with similar co-occurring psychosocial issues to those affected by CSE [2,3]. Mentoring, multi-systemic treatment/family

therapy and professional supportive therapeutic relationships have also been common interventions utilized amongst this cohort [2,3]. Additional CSE interventions include youth-focused relationship education programs, targeting the improvement of healthy relationship knowledge [3]. Meta-analytic data suggests curriculum focused on respectful relationships can improve healthy relationship knowledge and attitudes for youth in schools that engage in these programs [34]. Problematically, however, youth and children who are frequently assessed as 'at-risk' of, or exposed to CSE, are known to experience significant social disconnection from families and school, limiting access and exposure to these resources [28]. Other efficacious programs which provide direction for intervention are those designed to target sexual risk behaviours in adolescent females [35] warranting examination in the CSE context. Overall, research suggests the most successful programs in CSE intervention appear to be those that integrate psychoeducation (e.g., interpersonal and sexual safety knowledge) with skill building (e.g., emotion regulation skills), however few studies target both [2,36].

## 2.2. Emotion Regulation and CSE

CSE is well known to be associated with a wide range of psychopathologies and psychosocial vulnerabilities, including CSE [9,37,38]. Research indicates the link between abuse in childhood and psychopathology lies in the disruption of the development of adaptive emotion regulation processes [38,39]. Within this conceptual framework, emotion regulation is defined as the processes responsible for monitoring, evaluating, and modifying the expression and experience of emotion, to accomplish desired goals [26]. Emotion regulation can be assessed in several ways, with multidimensional approaches measuring 'trait' emotion regulation via self-report items, instructing participants to rate their average experience of emotion [40]. Trait-based emotion regulation measurement provides information about an individual's overall disposition or propensity within certain emotion regulation domains. In contrast, emerging emotion regulation research presents a state-based approach, whereby momentary or day-to-day aspects of emotion are measured, to accommodate the impact of situational factors, such as interpersonal experiences and cognitive processes which manifest themselves in specific emotional experiences (e.g., in the aftermath of trauma or loss, or losing behavioral control due to shame/guilt: [41,42]). Evidence-based recommendations indicate a need for measuring emotion regulation with both trait and state approaches [41].

Emotion regulation intervention is recognized as a transdiagnostic treatment approach that targets strategies and skills individuals utilize to regulate emotion, including awareness of and acceptance of emotion, the ability to control impulses and emotional intensity, and the use of flexible strategies befitting to a situational context [26,40,43]. Transdiagnostic approaches cut through a myriad of symptoms, focusing on one common denominator in psychopathology [44]. These interventions conceptualize mental health disorders as the result of core processes that underlie them, for example, rumination or emotional suppression [32]. In light of meta-analytic research that reports associations between emotion regulation difficulties and CSE [9] and evidence that implicates maladaptive emotion regulation as a risk factor for sexual re-victimization [22], incorporating emotion regulation intervention into programs that support this young cohort may be beneficial.

## 2.3. The Current Study: ERIC + YR Intervention

Considering existing literature recommendations to embed skill building and safe relationships psychoeducation into CSE intervention design [2,45], the present study piloted an 'Emotion Regulation and Impulse Control' (ERIC) [46] and safe relationships (Your Relationships (YR) [47]) intervention program called 'ERIC + YR', for young women affected by CSE. The aim of this study was to examine the effects of ERIC + YR on the outcomes of emotion regulation, psychological wellbeing, and safe relationship knowledge and behaviours, in a randomized case series trial. Specifically, it was hypothesized that young women who receive the ERIC + YR intervention would report improvements from

baseline (Phase A) to end of active intervention period (Phase B) across the following outcome variables:

I.   Self-reported emotion regulation (Difficulties in Emotion Regulation Scale) [40,48].
II.  Self-reported emotion regulation strategy use (ERSS) [49].
III. Self-reported psychological wellbeing (Outcome Rating Scale) [50].
IV.  Self-reported relationship safety (Relationship Safety Survey).

## 3. Materials and Methods

### 3.1. Setting

A community service organization dedicated to supporting young women affected by sexual exploitation in Melbourne, Australia served as the partner for recruitment and implementation of this study. Service provision included outreach for young women (12–25 years) affected by sexual exploitation, with a trauma-informed and relational approach.

### 3.2. Participants

Young women at the service were assessed for eligibility to participate by members of the research team. Inclusion criteria were: (1) aged between 16 and 18 years; (2) with a history of CSE; (3) who had the capacity to provide informed consent, OR if aged 16–17, had a legally acceptable representative to provide written informed consent; (4) who had a mobile smartphone with internet access; and (5) were fluent in the English language. Exclusion criteria were: (1) symptoms of active psychosis; (2) schizophrenia spectrum disorders; (3) acute crisis presentation (e.g., intoxication or withdrawal episode, severe depressive episode requiring hospitalization, domestic violence situation, housing crisis); and/or (3) intellectual disability or neurodevelopmental disorder if preventing ability to provide informed consent. Three young women enrolled in the study and completed baseline measures. Two young women completed the full study protocol, including baseline assessment, daily measures, all intervention sessions, and the postintervention session. The third young woman chose to withdraw from the study, due to service closure in lieu of the COVID-19 pandemic and a lack of time to finish the ERIC + YR program. All included participants had experienced a pre-existing history of CSE, prior to the age of 15 years (see Table 1 for sample characteristics).

**Table 1.** Demographic characteristics of the sample.

| Characteristic | Participant 1 | Participant 2 |
|---|---|---|
| Age | 18 | 18 |
| Sex | Female | Female |
| Sexual orientation | Heterosexual | Lesbian |
| Place of birth | Australia | Australia |
| Educational attainment | Year 10 | Year 10 |
| Current education engagement or employment | /No | Yes |
| Relationship status | Yes, Boyfriend | Yes, Girlfriend |
| **Experience of CSE** | **Yes, at <15 years of age** | **Yes, at <15 years of age** |
| Co-morbidities | No | PTSD * |
| Emergency department admission (last 12 months) | No | Yes, suicide attempt |
| Ever experienced homelessness | No | Yes, at age 15 |
| Engaged with foster care system | No | No |
| A parent themselves | No | No |
| Contact with the police in the past two weeks | No | Yes |

Note. PTSD * = Post Traumatic Stress Disorder. **Bold** denotes CSE experience <18 years of age.

### 3.3. Materials

The ERIC and YR interventions were integrated to be delivered together across eight sessions, to include both psychoeducation with a complementary emotion regulation exercise, and reflection to prompt daily repetition of new skills (see Table 2 for ERIC + YR domains and exercises).

**Table 2.** ERIC + YR Domains and Exercises Delivered During Intervention [51,52].

| Intervention Target | Worksheets | Exercises | Theoretical Basis |
|---|---|---|---|
| Human rights and needs | Human Needs | Psychoeducation to normalize safety as a human right and need for children and young people. Practice identifying needs that are fulfilled and those that are unmet. Reflection regarding what safety means. | Maslow's hierarchy of human needs [53]. National Children's and Youth Law Centre [54]. |
| Self-Care and Self-Compassion | 5 Self-Care Habits | Developing a self-care plan that involves reaching out to others, exercising, and sleeping well, being mindful, eating well and being kind to yourself and others. | DBT self-soothing [55]. Compassion Focused Therapy [56]. |
| Emotional literacy | Why should I regulate Dissecting your feelings | Psychoeducation and insight building into current patterns of emotion regulation A functional analysis of physical sensations, emotions, urges, cognitions, and behaviors during a chosen situation. | The Unified Protocol for the Transdiagnostic Treatment of Emotional Disorders [57]. A CBT based functional or chain analysis [58]. |
| Power differentials in relationships | Doing what I need (Blink, Think, Choice, Voice) Power and Control | A mnemonic for teaching young people about intelligent disobedience and how to apply it in relationships when disobeying is the right thing to do. Creating awareness of power differentials in the context of all sorts of safe and unsafe relationships. | Intelligent disobedience [59]. Duluth model of power and control [60]. |
| Consent | Consent on or offline | Psychoeducation regarding consent, encompassing discussions of verbal yes, no coercion and within the context of equality. | Respectful Relationships: Teaching and Learning Package [61]. |
| Behavioural Avoidance | Facing up to avoidance | Developing a behavioral experiment to engage in graded exposure to a situation that has been avoided. | A CBT based behavioral experiment [62]. |
| Acceptance | Allow space for all your feelings | Identification of emotions that are currently avoided and a graded exposure plan to experience them a little bit each day. | CBT for emotional disorders [58]. |
| Interpersonal skill–boundaries | Boundaries Insight Cards | Using a set of visual images depicting safe and unsafe relationships to reflect on boundaries on and offline. Differentiating between no boundaries, uncertain boundaries, and healthy boundaries. | Interpersonal effectiveness skills from DBT [55]. Respectful Relationships: Teaching and Learning Package [61]. |
| Distress Tolerance | Shake off feelings | Develop a behavioral plan that allows cognitive disputation of thoughts and behaviors that perpetuate a negative emotional state. | Opposite action, distress tolerance skill from DBT [55] and CBT based behavioral experiments [62]. |
| Mindfulness | Mindful breathing Mindful lean | Using the spotlight of attention to aid mindful breathing Using the feet and toes to help check in with the present moment. | Three-minute breathing space [63]. Physical sensations in mindfulness [64]. |
| Sexual violence awareness and reporting | Sexual Exploitation | Exploring what constitutes CSE, watching a 3-min video of a victim-survivor's experience, exploring accessible support services and reporting both online and offline. | Preventing online child sexual exploitation [65]. Respectful Relationships: Teaching and Learning Package [61]. |
| Values and identity | No matter how you feel, do what matters to you | A metaphor of passengers in a minivan to represent cognitions and emotions that are commonly avoided and a road trip to represent value-based action. | Passengers on a bus metaphor from ACT [66]. |

### 3.3.1. Emotion Regulation Impulse Control Program

ERIC [46] is a transdiagnostic and modular intervention which targets the acquisition of adaptive emotion regulation skills such as problem solving, acceptance, mindfulness and cognitive reappraisal, and reduction of avoidance, rumination and suppression. The intervention is described in depth in the intervention manual [34] and all intervention elements are outlined in Table 2. ERIC is designed to promote healthy emotional and social development for vulnerable young people by cultivating helpful emotion regulation and impulse control skills. ERIC has been found to reduce emotion dysregulation and distress in a case series of vulnerable young people [51]; and in a larger effectiveness pilot [52] found to be acceptable and feasible to deliver to vulnerable young people across a broad range of youth services, including Alcohol and other Drug (AOD), primary youth mental health, youth justice, community services and primary care settings.

### 3.3.2. Your Relationships

Your Relationships (YR) [47] is a psychoeducational intervention aimed at promoting safe relationships amongst 12-to-25-year-old young people. YR is built on the principles of cognitive-behavioral therapy empowering young people to explore thoughts and feelings regarding safe relationships, power and control, sexual health, consent, self-worth, sexting, gender expectations and sexual abuse. YR consists of exercises that target the application of interpersonal skills within a therapeutic relationship and utilize a pack of 20 visual cards to elicit conversation from a strengths-based framework (see Table 2). YR was developed by a youth community service that supports young women affected by CSE, from a trauma-informed framework, and in consultation with survivors of sexual exploitation. YR was developed through an iterative process including involvement from young people, adult survivors, expert reviewers and focus groups, and piloted with practitioners to ensure its utility and feasibility.

### *3.4. Measures*

### 3.4.1. Emotion Regulation Strategies Scale

The Emotion Regulation Strategies Scale (ERSS) [49] captures the use of adaptive and maladaptive emotional regulation skills in daily life. Several strategies are measured including acceptance, behavioral avoidance, distraction, suppression, reappraisal, reflection, and rumination. Participants read items such as 'I accepted how I was feeling' and 'I avoided the situation that led to my feelings' and are asked to rate how frequently they use these strategies from 1 (never) to 5 (almost always). A total score is accumulated to represent total emotion regulation strategies utilized each day, with higher scores representing improved use of adaptive strategies. The ERSS was selected due to its alignment of trait and daily emotion regulation strategy use, face validity, complementary outcomes with the ERIC + YR domains, and its brief nature maximizing utility in a daily measurement while reducing participant burden.

### 3.4.2. The Outcome Rating Scale

The Outcome Rating Scale (ORS) [50] is a brief outcome measure designed to measure psychological well-being and progress during and after intervention in clinical practice. The ORS invites individuals to indicate their current level of distress and functioning across four items which represent discrete life domains; including (1) personal or symptom distress (individual wellbeing), (2) interpersonal wellbeing (how an individual is getting along in the family and close relationships), (3) social wellbeing (measuring satisfaction with school/work and relationships outside the home) and (4) overall (general sense of well-being). Participants self-report their perceived level of wellbeing by scoring between 0 and 10, with 0 out of 10 representing distress and 10 out of 10 indicating the highest level of functioning and satisfaction. Scores below 25 indicate clinically significant symptoms. The ORS is a reliable and valid measure for young people and adults, with high utility and only requiring on average one minute to complete.

### 3.4.3. Difficulties in Emotion Regulation Scale Short Form

The Difficulties in Emotion Regulation Scale Short Form (DERS-16) [48] is a theoretically driven self-report measure of an individual's typical level of emotion dysregulation across six trait-based dimensions. The DERS consists of 16 self-report items which measure emotion dysregulation dimensions of nonacceptance of negative emotions (3 items), inability to engage in goal-directed behaviours when distressed (3 items), difficulties controlling impulsive behaviours when distressed (3 items), limited access to emotion regulation strategies perceived as effective (5 items), lack of emotional clarity (2 items: Bjureberg et al., 2016). The DERS uses a Likert scale from (almost never) to 5 (almost always), with a total score ranging between 16 and 80, and higher scores reflecting greater levels of dysregulation. Total DERS scores were summed to compare pre and post data. The DERS-16 is highly correlated with the original 36-item version and has adequate internal reliability comparable to the original measure ($\alpha$ = 0.95; [40]).

### 3.4.4. Relationship Safety Survey

The Relationship Safety Survey (RSS: developed for this study) was developed for this study based on past research, measuring perceptions and behaviours associated with sexual coercion, sexual exploitation, online grooming and equal relationships. To reduce the burden on participants, only one item of the 15-item RSS are utilized in the daily outcomes to measure relationship safety ('When I feel unsafe in a relationship, I have strategies to avoid this') and knowledge of relationship safety ('It's ok for someone to trick, threaten or force me into doing something sexual I do not want to if it does not physically injure me'). Respondents rate their responses on a Likert scale from 1 (never) to 5 (always). The complete scale (15 items) was used in the pre and post data collection.

### 3.4.5. Interpersonal Competence Questionnaire-Revised

The Interpersonal Competence Questionnaire-Revised (ICQ-R) [67] consists of 8-items from the original ICQ-R. The ICQ-R measures interpersonal skills across three domains, asserting influence, self-disclosure and conflict-Resolution. Respondents self-rate their confidence in various competencies in specific social situations on a scale from 1 to 5. The score for each scale is the average scale score. The chosen ICQ-R subscales will measure the young person's capacity to say no to confronting, uncomfortable or unreasonable requests in relationships, their capacity to seek help or allow intimacy in healthy relationships, and conflict management skills. The ICQ-R has high reliability and validity ($\alpha$ = 0.86: [41]).

### 3.4.6. Conflict in Adolescent Dating Relationships Inventory Short Form

The Conflict in Adolescent Dating Relationships Inventory Short Form (CARDIS) [68] is a ten-item bidirectional (victim/perpetrator) questionnaire which measures adolescent dating violence across five domains of physical abuse (e.g., 'I slapped my partner or pulled their hair'/'my partner slapped me or pulled by hair'), threatening behaviour (e.g., 'I threatened to hurt my partner'/'my partner threatened to hurt me'), sexual abuse ('I forced my partner to have sex when he/she did not want to/my partner forced me to have sex when I did not want to'), relational abuse (e.g., 'I tried to turn my partners friends against them'/'my partner tried to turn my friends against me') and verbal/emotional abuse '(I insulted my partner with put-downs'/'My partner insulted me with put-downs)'. Response choices for each item are defined as never (this has never happened), seldom (this has happened 1–2 times), sometimes (this has happened 3–5 times) and often (this has happened more than 6 times). The CARDIS has previously been validated with high school students and at-risk youth, presenting strong validity and reliability ($\alpha$ = 0.85: [42]).

### 3.5. Design

A randomized single-case series A-B phase design was used to test the effects of ERIC + YR on emotion regulation strategies and psychological wellbeing across repeated measurements. There were two phases in the study: Phase A-Control (completion of baseline and

daily measures) and Phase B-Intervention (8-session ERIC + YR activities + completion of daily and follow-up measures). Participants were assigned a randomized start-point where they switched from the control phase to the intervention phase. In this study, measures were taken daily (7:30 p.m.) over the course of 40 days. The control phase lasted for a minimum of 20 data points/days to establish a baseline for the outcome measures. Dependent on what start-point participants were randomized to, they then switched from control to intervention phase and completed the same daily and follow-up measures while undertaking the 8-session ERIC + YR intervention, lasting from 40 data points/days to a maximum of 50 data points/days. The outcomes were daily emotion regulation strategies, psychological wellbeing, and safe relationship behaviors. In addition to the daily outcomes' measures, baseline and post intervention outcomes included a multidimensional measurement of emotion regulation, interpersonal competency, safe relationship knowledge, self-reported sexual safety behaviours and sexual risk behaviours.

### 3.6. Procedure

This study was approved by the relevant university human research ethics committee. ERIC + YR was offered as an adjunct intervention to young people who were at varying stages (six months to 2 years) of their participation and support by the community service. Service users were informed of the study through an advertisement card provided by their practitioner, flyers posted around the service, and verbally by their practitioner. Rolling recruitment occurred from January 2020 to April 2020; however, recruitment was ceased at the implementation of a state-wide lockdown in lieu of the COVID-19 pandemic. Young people were given the ethics approved participant information sheet for their consideration. Written informed consent was obtained by the person who conducted the informed consent discussion (i.e., the researcher). After providing informed consent and completing the pre-intervention period (Phase A), participants received a $15 gift card. Participants were randomized to a start-point and asked to engage in monitoring their emotion-regulation strategies and psychological wellbeing over the next 20 days. Dependent on their start-point, they switched from Phase A to Phase B and completed the same outcome measures alongside ERIC + YR. Participants received email links to mobile phone app surveys daily, at 7:30 p.m. for 3 weeks to collect 20 baseline data points (Phase A). Each survey asked questions regarding emotion regulation strategy use, (McMahon & Naragon-Gainey, 2019), psychological wellbeing (Miller, Duncan, & Brown, 2003), and relationship safety over the last 12 h. Surveys took 3–5 min to complete. At completion of Phase B, participants received a $35 gift card.

Practitioners from the service attended a 1-day workshop, providing an overview of the research requirements and training to implement the ERIC + YR intervention. Learning objectives included being able to explain study participation to young people and competently administer the ERIC + YR materials to young people. The workshop addressed skills in delivering the ERIC + YR intervention, accessing and navigating the intervention manual and clinical materials, and applying the ERIC + YR tools to working with young people. The learning methods included didactic, role-plays, case discussions and small group activities. The facilitators were the authors of the ERIC + YR interventions, Dr. Kate Hall (clinical psychologist) and Jessica Laird (registered nurse and provisional psychologist conducting research as part of a doctorate in clinical psychology).

### 3.7. Data Analysis
3.7.1. Statistical Significance

A permutation test was used to assess statistically significant differences between Phase A and Phase B on scores of the daily measures (ORS and ERSS). These indicate the probability of change occurring at the specific nominated randomized start point for the participant in the context of all possible start-points. and provides corresponding *p*-values and standardized effect sizes. A randomization test with this design was powered to indicate whether standardized mean score differences of 0.6 or greater between

Phase A and Phase B represent statistically significant differences, with an alpha of 0.05 and power > 0.80. Randomization for start-point allocation and statistical analyses were conducted using EXPRT (Excel® Package of Randomization Tests): Statistical Analyses of Single-Case Intervention Data (Version 4.2, November 2021: [69]). The *p*-values and effect sizes derived from individual participants were meta-analyzed using the additive method [70]. To estimate the strength of effects Cohen's *d* was used.

### 3.7.2. Clinical and Reliable Change

To detect both reliable and clinically significant change across the intervention (from phase A to B) the reliable change index (RCI) [71] and clinical change score [72] were utilized, calculated with the mean scores of the DERS-16 and ORS. Beyond statistical significance, the RCI and clinical change scores provide additional information regarding the magnitude of change produced by intervention, and whether this change is reliable. The RCI method of analysis is designed to determine whether an individual's observed change was greater than the change that would be expected due to change and measurement error (i.e., reliability of the measure). Specifically, the RCI formula by Christensen and Menfoza [45] was used, which accounts for unreliability in both pre- and postscores (see Formula (2) from Hageman & Arrindell, 1992, [73] for further details). This formula is considered more conservative and methodologically sound than the original RCI formula [47]. Specific reliabilities and standard deviations from normative populations that were used to derive the RCI for each variable are presented in Table 3. Given the RCI is a z-score, RCI values N |1.96| were deemed to indicate statistically significant change. A clinically significant change was analyzed to determine whether an individual's score at post treatment was closer to the mean score of a healthy population rather than the mean score of a clinical population. Specifically, the method c approach was used as outlined in Jacobson and Traux [72] (1991) and normative data from published studies (Table 3). The Daily Emotion Regulation Strategies Scale has no normative data currently, therefore reliable and clinically meaningful change was unable to be calculated for this scale [49].

**Table 3.** Data used to compute reliable and clinically significant change.

|  | DERS-16 | ORS |
|---|---|---|
| Non-Clinical mean, M0 | 33.57 | 28.00 |
| Clinical mean, M1 | 57.00 | 19.60 |
| Non-clinical SD, S0 | 13.14 | 6.80 |
| Clinical SD, S1 | 13.05 | 8.70 |
| Reliability, rsx | 0.94 | 0.93 |
| [a] Standard error of measurement, SE | 3.32 | 2.30 |
| [b] Cut off for clinically significant change | 45.32 | 23.29 |

Note: DERS-16 = Difficulties in Emotion Regulation Scale [48]; ORS = Outcome Rating Scale [50]; [a] Derived using Formula (2) from Hageman and Arrindell [73]; [b] Derived using method c approach from Jacobson and Traux [72].

## 4. Results

While a complete data set was collected for the two included participants, it should be noted that due to the impact of COVID-19 (e.g., service closure) the final survey responses and measures were fast-tracked over a fortnight. Following randomized start points, P1's total exposure to the program was eight weeks and P2's was six weeks. Table 4 presents the descriptive statistics and effect sizes for the daily measures for participant one (P1) and participant two (P2). Graphical representations of changes between the phases over time are shown in Figures 1–5. No variance was found across the self-reported relationship safety measure for P1 (e.g., all data points were reported as 4 out of 5 on the Likert scale) therefore these were excluded from the AB analysis, as responses without variance result in the inability to estimate missing data, standard deviation or effect size. P2's relationship safety knowledge indicated a small mean improvement; P1 and P2 had a small improvement across mean daily emotion regulation strategies; and psychological wellbeing was relatively

static for P1, and increased from baseline to intervention phase for P2. However, no results were statistically significant.

**Table 4.** Descriptive statistics, *p*-values and effect sizes for outcomes (daily measures).

| Participant | Control Mean | ERIC + YR M | Control Phase SD | Cohen's d |
|---|---|---|---|---|
| Emotion Regulation Strategy Use (Daily ERSS Total Score) | | | | |
| P1 | 16.83 | 17.95 | 1.87 | 0.60 ns |
| P2 | 23.84 | 24.50 | 4.48 | 0.15 ns |
| Psychological Wellbeing (ORS Total Score) | | | | |
| P1 | 29.10 | 27.62 | 1.96 | −0.75 ns |
| P2 | 15.17 | 19.16 | 8.95 | 0.45 ns |
| Relationship Safety Knowledge (RSS item) | | | | |
| P2 | 2.53 | 3.10 | 1.07 | 0.53 ns |

Note: P = participant; SD = standard deviation; ERSS = Emotion Regulation Strategies Scale; ns = not statistically significant; ORS = Outcome Rating Scale; RSS = Relationship Safety Scale.

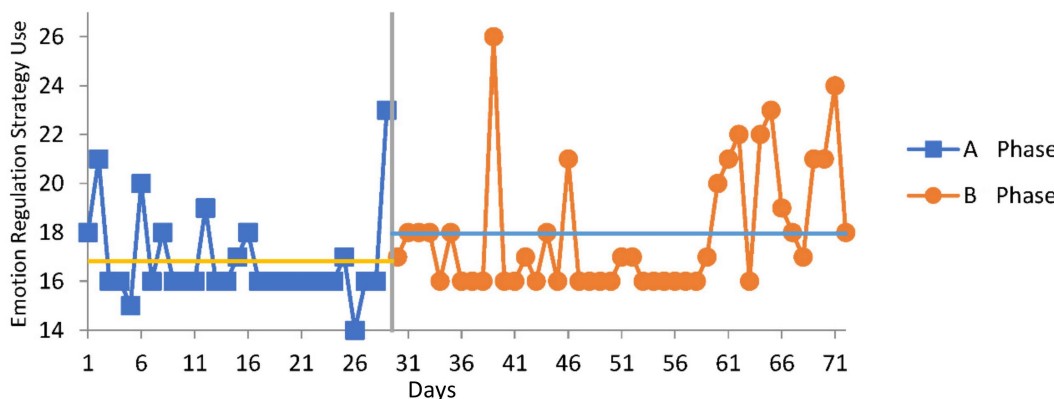

**Figure 1.** Emotion Regulation Strategy Use changes from phase A to B for P1.

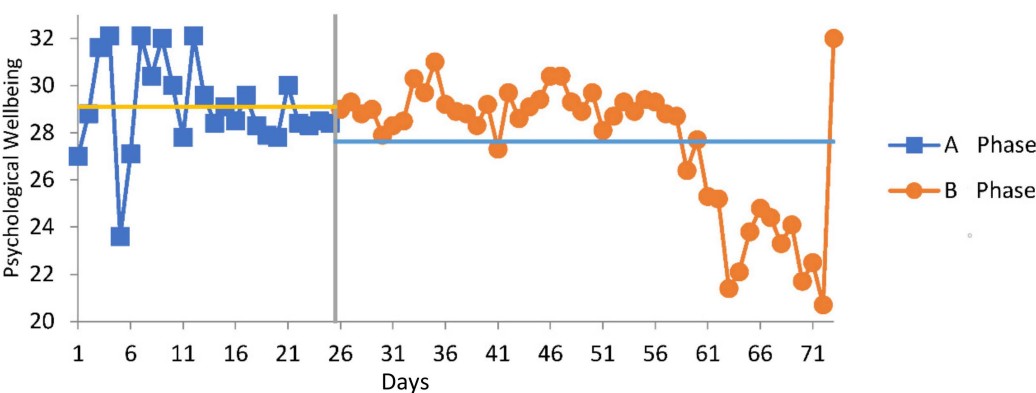

**Figure 2.** Psychological Wellbeing changes from phase A to B for P1.

Table 5 presents pre and post intervention raw scores across listed outcome measures for each participant. P1's emotion regulation strategy use remained static across the 72 days; however, their emotion dysregulation symptoms indicated a clinically significant and reliable increase, contrary to what was hypothesized. P1's results indicated no other significant change. P2 indicated stable emotion regulation strategy use and reduced emotional dysregulation symptoms; however, neither of these were statistically significant. P2 reported reliable and clinically meaningful improvement in their psychological wellbeing as indicated by the outcome rating scale. No other outcome measures resulted in a significant change between baseline and intervention scores.

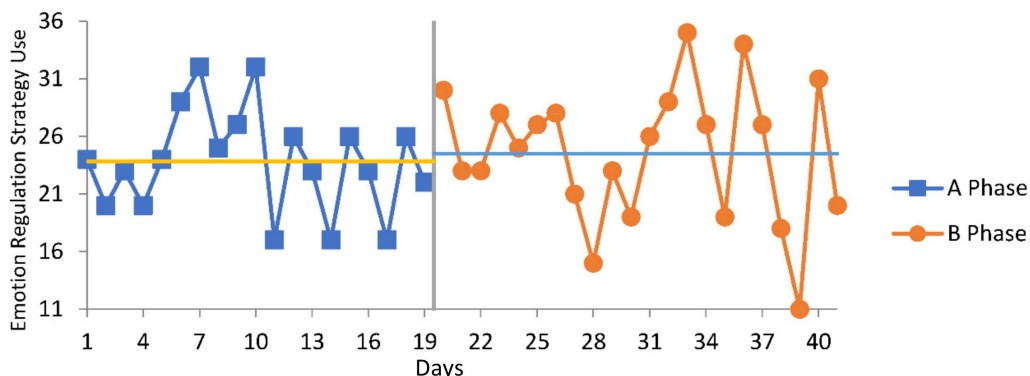

**Figure 3.** Emotion Regulation Strategy Use changes from phase A to B for P2.

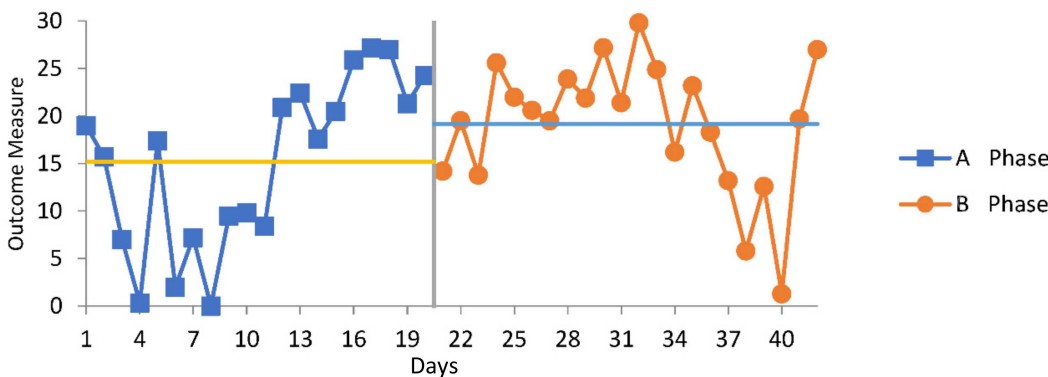

**Figure 4.** Psychological Wellbeing changes from phase A to B for P2.

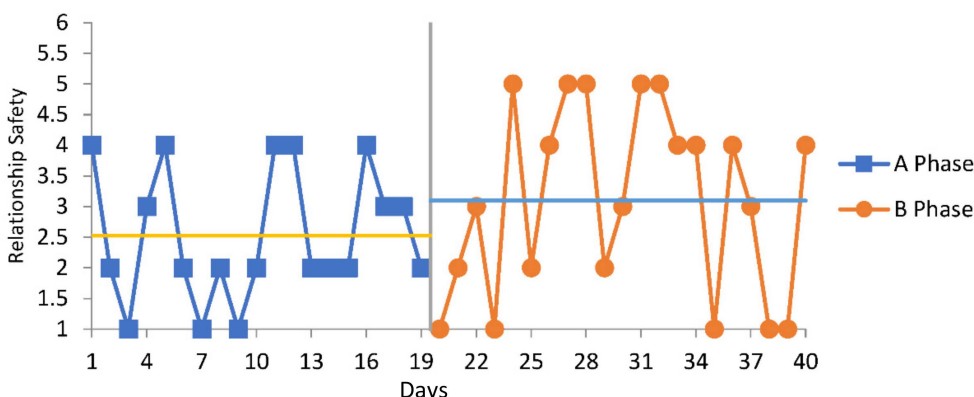

**Figure 5.** Relationship Safety changes from phase A to B for P2.

**Table 5.** Pre and post treatment scores.

|  | Participant 1 (P1) | | Participant 2 (P2) | |
| --- | --- | --- | --- | --- |
|  | **Pre** | **Post** | **Pre** | **Post** |
| Daily emotion regulation strategy use (ERSS) 1 | 18 | 18 | 24 | 20 |
| Emotion dysregulation symptoms (DERS-16) | 19 | 34 */# | 56 | 40 |
| Psychological wellbeing (ORS) | 27 | 32 | 19 | 27 */# |
| Interpersonal competence (ICQ) | 54 | 61 | 71 | 77 |
| Relationship safety knowledge and behaviour (RSS) | 36 | 41 | 34 | 44 |
| Conflict in dating relationships (CARDIS) | 0 | 2 | 0 | 5 |

Note: 1 = Raw scores reported only, without clinical or reliable change calculated due to a lack of normative data available. * Reliable significant change at $p < 0.05$; # Clinically significant change; P = participant number; DERS = Difficulties in Emotion Regulation Scale; ORS = Outcome Rating Scale; ICQ = Interpersonal Competence Scale; RSS = Relationship Safety Scale; CARDIS = Conflict in Adolescent Dating Relationships.

## 5. Discussion

Despite the extreme violence embedded within CSE, there remains a lack of literature in the current evidence base regarding existing intervention. Recent research indicates interventions implemented for young people affected by CSE are often tailored towards other co-occurring social issues (e.g., drug and alcohol misuse [2]) or hardly exist at all, with a systematic review reporting only 21 studies investigating CSE interventions globally from 1991 to 2015 [3]. Furthermore, existing intervention studies pertaining to individuals affected by CSE lack rigorous research methods and fail to provide robust outcome data [3]. Therefore, to advance the CSE evidence-base, this study aimed to inform future intervention research by designing and piloting a quasi-experimental combined emotion regulation and relationship safety program, and investigating the outcomes of emotion regulation, psychological wellbeing and safe relationship behaviours, in a sample of young women affected by CSE. The case series consisted of two young women who experienced CSE before the age of 15 years who were currently engaged with a community support service for young women affected by sexual violence. Assessments were administered (a) at baseline pre-intervention, (b) daily during the eight-session intervention phase across three to six weeks, and (c) at follow-up a few days to one week post completion of ERIC + YR. The results of the program and implications for future researchers are discussed.

### 5.1. Emotion Regulation

Overall, the day-to-day aspects of emotion regulation which accommodates situational and interpersonal experiences across a 12-h period, remained relatively static for both participants; therefore, contrary to our hypothesis, no statistically or clinically meaningful differences between baseline and intervention scores were detected. This may reflect a lack of emotion regulation skill acquisition in the short period of time; the need for a higher dose of intervention sessions to reach improvement at a statistically significant threshold; or the need for longitudinal research to detect gain across emotion regulation skills following intervention delivery; or due to confounding factors in the participants lives, unmeasured and beyond the program itself. In a 2021 study, the ERIC program was implemented for 12 weeks alongside care as usual for 79 young people, and a significant reduction in emotion dysregulation was reported [52]. Furthermore, in Hall et al. (2021) [37] participants were followed-up approximately six weeks post intervention. The current study however was truncated due to the COVID-19 pandemic, with participant exposure to the program ranging from three to six weeks, and follow-up measures within days of the final dose of ERIC + YR. Therefore, it is possible the ERIC + YR program was not able to reach a therapeutic dose to detect changes over time. According to Figure 1, P1's emotion regulation strategies visually appeared to be trending upward, indicating a possible increase in emotion regulation strategy use, yet no statistical significance was reached. It is possible that further exposure to the ERIC + YR program or a longer period of time between intervention and follow-up may have found clinically relevant changes in emotion regulation strategy use.

Contrary to P1's overall improvement across all measures (though not statistically significant) they exhibited an unexplained clinically significant increase in emotion dysregulation symptoms in the follow-up measures. P1's low variability and lack of significant change across their daily emotion regulation would suggest that no major interpersonal experiences, or situational factors, escalated their emotional experience across the intervention; however, it is possible that the pandemic and threat of closure to the service influenced the young person's emotional dysregulation on the day of follow-up. While the service was intended to provide care as usual pre and post intervention, the pandemic resulted in the removal of face-to-face support, with the provision of telehealth only. Research has shown that the impacts of COVID-19 has amplified violence against women and children and consequently reduced mental health for women and children specifically [74]. Considering the sample characteristics (i.e., young women), alongside pre-existing trauma and violence histories of those affected by CSE, it is plausible that the pandemic increased risk

and psychosocial stressors for participants towards the end of intervention. Where the current study used quantitative methods, future studies may benefit from a mixed methods approach, collecting longitudinal and qualitative data to examine the impact of contextual factors on the participants emotional experiences.

### 5.2. Psychological Wellbeing

As hypothesized, the psychological wellbeing of P2 indicated clinically and meaningful improvements post intervention. This result means that when compared to the beginning of the ERIC + YR program, the young woman perceived improved overall psychological wellbeing after completion of the intervention. Specifically, P2 self-reported perceived reduction in psychological distress, improved interpersonal and social relationships, improved satisfaction in social roles outside of the home (e.g., work/school) and improved psychological wellbeing overall. However, future research would benefit from examining the specific interventional component that may have contributed to the change, which may vary from the therapeutic relationship with the practitioner, the intensity of contact between the individual and practitioner, to the psychoeducation or emotion regulation interventional skills themselves. No significant findings were reported for P1.

### 5.3. Safe Relationships Psychoeducation

Changes in self-reported scores of safe relationship knowledge or behaviors were undetected. It is possible that the limited data variance in the range of scores reflects a degree of response bias. This may be evidenced by both uniform responses (e.g., over 70 data points the respondent chose the same answer such as 'somewhat disagree' present on the relationship safety survey) and neutral responses (e.g., responses remaining neutral every time, present across other measures). All relational and interpersonal measures were placed at the end of the daily measure and the pre/post measures, potentially indicating participant fatigue or attentional burden may have contributed to static responding. It is also possible that these results were impacted by sudden changes in the social environment due to COVID-19, and external forces disrupting follow-up.

### 5.4. Implications

Findings from this pilot have implications for program developers, frontline practitioners and researchers, who aim to create evidence-based interventions which support young people affected by CSE. Research shows that sexually exploited youth benefit from multi-systemic interventions which incorporate family, community, education and protective systems, build rapport, provide psychoeducation of structural violence, manage risk and safety [75] and treat psychological symptoms of trauma [3]. In addition, current literature suggests intervention design for young people affected by CSE would benefit from increased rigor in research design and partnerships with frontline programs [3]. Therefore, key strengths of the ERIC + YR program design present in its unique ability to: (a) implement a complete evidence-based transdiagnostic intervention for young women affected by CSE; (b) alongside rigorous and empirical research; and (c) to provide this support to young people who remain engaged and safeguarded by community service, to maintain continuity of care, a therapeutic relationship, monitor risk and provide psychosocial supports. While literature suggests young people affected by CSE are a notoriously high-risk cohort with complex needs, who are often reported to be challenging to engage in both research, education and intervention [76], this novel research methodology was able to be fully piloted.

Increasing community awareness of CSE and improving training for frontline responders form a part of the national action plan to safeguard children and young people in many countries [6]. An additional advantage of integrating a program such as ERIC + YR into community services is the training of frontline practitioners. Pre-intervention the ERIC + YR program provided the community service with psychoeducation regarding trauma dynamics associated with CSE, issues of power, consent, sexual abuse stigma-

tization, sexual health and the safe use of digital mediums; as well as equipping staff with evidence-based interventional skills which cultivate healthy social and emotional development in young people. Practitioners reported feeling 'empowered' by ERIC + YR training, as they were able to access the intervention content and resources permanently. Practitioners reported the tools were 'easy to use' and 'enjoyable' to implement. Importantly the young women were also observed by the practitioners to be 'engaged' with the content. This may be further evidenced by the young women's consistency in completing daily measures in the smartphone app for up to six weeks. While the current study builds on the body of current CSE research, it is important to explore learnings and limitations to offer insight for future interventional design.

### 5.5. Limitations

The interruption of the recruitment phase (secondary to COVID-19 and community service closure) resulted in a small sample size; therefore, generalizability is not possible, and findings should be interpreted with caution. Furthermore, the fast-tracked implementation of the ERIC + YR program towards the end of data collection, and the collection of follow-up data within days of the final ERIC + YR session, may have limited the detection of therapeutic impact of the program. Further research may benefit from exploring the longitudinal outcomes post intervention. In addition, it is likely that COVID-19 increased the social and individual risks faced by the young people in these case studies; however, these factors were unable to be measured due to a lack of qualitative data. Suggestions for overcoming the limitations for future research and practice are discussed.

### 5.6. Future Directions
#### 5.6.1. Measures

Re-implementation of the program would benefit from incorporating multi-informant approaches to assessment and data collection (e.g., self-report and practitioner-rated measures). This may highlight if there are differences between self-perceived and practitioner-perceived changes within wellbeing, emotional regulation and relationship safety. Furthermore, measures that capture the strength in the therapeutic alliance may allow for the control of this effect and/or detect if is a confounding variable.

#### 5.6.2. Mixed Methods

Embedding a qualitative component (e.g., thematic analysis of interview data) may provide needed insight regarding intervention feasibility and user experience (practitioner and young person), response biases if present, and gathering information beyond individually focused factors to better understand ecological influences (e.g., pandemic, or other environmental variables). Qualitative data may also allow for examination of differences between participant results, for example, why one young woman experienced improved psychological wellbeing, but another did not. Furthermore, while treatment manuals were provided to practitioners, qualitative data may deepen understanding of adherence to the treatment protocols.

#### 5.6.3. Sample

While this unique study design was adequately powered to detect statistical change, participant recruitment was minimized in response to pandemic-related lockdowns; a larger sample would have provided greater generalizability of results. Furthermore, participants had engaged with the support service for varying amounts of time prior to the ERIC + YR program being delivered, therefore, it is recommended that future studies incorporate a larger sample for generalizability, and control for pre-intervention existing emotion-regulation skills.

### 5.6.4. Therapeutic Dose and Follow Up

It is possible that further exposure to the ERIC + YR program, or a longer time between intervention and follow-up may have found clinically relevant changes in emotion regulation strategy use. In a successful implementation of the ERIC program which was implemented for 12 weeks alongside care as usual for 79 young people, and follow-up at approximately six weeks post intervention a significant reduction in emotion dysregulation was reported [52]. The current study however was truncated due to the COVID-19 pandemic, with participant exposure to the program condensed into three to six weeks, and follow-up measures within days of the final dose of ERIC + YR. Several studies report treatment gains can continue to improve at 6 and 12 months post intervention for sexual abuse survivors [77]. Therefore, future studies may benefit from extending the intervention dose and/or collecting longitudinal data to detect changes in outcomes over time.

### 6. Conclusions

In conclusion, the current study is an initial step in highlighting how a rigorous empirically driven intervention could be delivered as an adjunctive treatment for youth affected by CSE. This study offers some evidence that suggests programs such as this can increase the overall psychological wellbeing of young women affected by sexual violence. While research consistently reports that samples of young people and children who have experienced CSE are notoriously difficult to engage and retain in intervention programs, this novel methodological design of ERIC + YR may offer direction for future interventive research. With strengths, limitations and learnings discussed, the current study expands the evidence base which seeks to advance trauma-informed and recovery-oriented interventions for youth affected by CSE.

**Author Contributions:** Conceptualization, J.J.L.; D.J.H.; K.H. and B.K.; Methodology, J.J.L.; D.J.H.; K.H.; B.K.; S.M.; Formal Analysis, J.J.L. and D.J.H.; Writing—Original Draft Preparation, J.J.L. and D.J.H.; Writing—Review and Editing, J.J.L.; D.J.H.; S.M. and K.H.; Supervision, B.K. All authors have read and agreed to the published version of the manuscript.

**Funding:** This research received no external funding.

**Institutional Review Board Statement:** The study was conducted according to the guidelines of the Deakin Higher Research Ethics Committee and approved by the Institutional Review Board of Deakin University of Melbourne, Australia on the 3 September 2019 (Application title: DUHREC ERIC + YR Emotion Regulation, Impulse Control + Your Relationships).

**Informed Consent Statement:** Informed consent was obtained from all subjects involved in the study.

**Data Availability Statement:** Data is contained within the article.

**Conflicts of Interest:** The authors declare no conflict of interest.

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
