# Peer review of "A Single-Case Series Trial of Emotion-Regulation and Relationship Safety Intervention for Youth Affected by Sexual Exploitation"

_psych, doi:10.3390/psych4030037_

Round 1

Reviewer 1 Report

This is an interesting manuscript on Child sexual exploitation (CSE). A randomised single-case series design was used to test the effects of ERIC+YR on emotion 19 regulation strategies, psychological wellbeing, relationship safety knowledge and behaviours, 20 across repeated measurements for young women affected by CSE. Evidence-base implications are listed as result of the current findings.

This work has many strengths, especially in the way the results are presented and designed. However, I feel that the final part does not match these strengths. I hope that my comments can help to strengthen the present work. 

The conclusions should be rewritten to try to focus directly on the results. They are too general and not directly related to the proposed study. I suggest the same recommendation for the discussion, even if the presentation of current results is much better.

On the other hand, the hypotheses are discussed, but they are not clear and operational prior to the design.

Author Response

Response to Reviewer 1 Comments

Point 1: This work has many strengths, especially in the way the results are presented and designed. However, I feel that the final part does not match these strengths. I hope that my comments can help to strengthen the present work. 

Response 1: Thank you kindly for your encouragement. This is an important area of research and we are so pleased to be able to contribute to the field. We hope to continue the research into evidence-based pathways out of exploitation for young people who need support.

Point 2: The conclusions should be rewritten to try to focus directly on the results. They are too general and not directly related to the proposed study. I suggest the same recommendation for the discussion, even if the presentation of current results is much better.

Response 2: Please find a re-written discussion and conclusion to telescope on the results more closely.

Point 3: On the other hand, the hypotheses are discussed, but they are not clear and operational prior to the design.

Response 3: Thank you for highlighting the need for clearer hypotheses. While these were developed in the protocol a priori to ethics and implementation, they were removed from the original MS for readability. They are now in the MS on page 5, lines 244 onwards.  

Reviewer 2 Report

Papers with small samples, null responses and quite exploratory are very hard to get published. In my view the paper needs a rewrite to narrow the literature review to the specific context (step away from international data as this has many complicating social contextual issues and does not add to the specific situation where this study was done). I have made several comments on the pdf attached. 

In the discussion, I would like to see some clarity on the lessons for future researchers - looking back what would you do differently? Were your measures good? Was the intervention really delivered as planned given the sudden impact of COVID which may add to the null results? The real goal, in my view, with null papers is helping future researchers. 

Thank you for the willingness to take a risk on submitting this paper.

Author Response

Response to Reviewer 2 Comments

Point 1: arrow the literature review to the specific context (step away from international data as this has many complicating social contextual issues and does not add to the specific situation where this study was done). 

Response 1: Thank you kindly for your very helpful and insights and feedback. We have arrowed the literature review to highlight data associated with CSE in the Western economies. Please find re-written sections from page 2 line 73 onwards in track changes.

Point 2: In the discussion, I would like to see some clarity on the lessons for future researchers - looking back what would you do differently? Were your measures good? Was the intervention really delivered as planned given the sudden impact of COVID which may add to the null results? The real goal, in my view, with null papers is helping future researchers. 

Response 2: We agree with the reviewers response. We have re-written our discussio and conclusion n to better orient future researchers to improve intervention efforts for young people affected by CSE. Tracked changes can be found throughout this section. Implications (Section 5.4 from line 542) and Future Directions (Section 5.6) are particularly altered.  

We have also paid attention to each of your comments throughout the MS, and amended accordingly. We found your feedback thoughtful and relevant. Thank you.

Round 2

Reviewer 1 Report

Thank you for addressing my comments

Author Response

No further changes.

Thank you to reviewer 1 for their contributions.

Reviewer 2 Report

Thank you for your attention to the revisions. The paper is much stronger. As you will see in my comments on the attached there are two areas in the revisions that I am suggesting some minor revisions.

Author Response

Response to Reviewer 2 Comments

Point 1: Re-phrasing line 84 to evidence harms associated with online and offline CSE.

Response 1: Thank you for your suggestion to clarify this evidence. We agree and have re-written the sentence on line 84 and 85.

Point 2: Line 128: reviewer recommended we expand on why children do not disclose CSE/CSA and incorporate supportive data from a literature review. 

Response 2: This reference was helpful in expanding data associated with CSE/CSA non-disclosure and delays. Please see the amended paragraph from line 128.

Thank you again for your considered feedback and time. We are grateful for your input.

Best regards.
